# Electric-Field-Induced Phase Transformation and Frequency-Dependent Behavior of Bismuth Sodium Titanate–Barium Titanate

**DOI:** 10.3390/ma13051054

**Published:** 2020-02-27

**Authors:** Kai-Yang Lee, Xi Shi, Nitish Kumar, Mark Hoffman, Martin Etter, Stefano Checchia, Jens Winter, Lucas Lemos da Silva, Daniela Seifert, Manuel Hinterstein

**Affiliations:** 1Institute for Applied Materials, Karlsruhe Institute of Technology, 76131 Karlsruhe, Germany; lucas.silva@kit.edu (L.L.d.S.); daniela.seifert@kit.edu (D.S.); manuel.hinterstein@kit.edu (M.H.); 2School of Materials Science and Engineering, UNSW Sydney, Sydney 2052, Australia; xi.shi@unsw.edu.au (X.S.); nitish.kumar@unsw.edu.au (N.K.); Mark.Hoffman@unsw.edu.au (M.H.); 3Deutsches Elektronensynchrotron DESY, 22607 Hamburg, Germany; martin.etter@desy.de; 4European Synchrotron Radiation Facility ESRF, 38043 Grenoble, France; stefano.checchia@maxiv.lu.se; 5MAX IV Laboratory, Lund University, 22100 Lund, Sweden; 6Department of Physics, University of Siegen, 57068 Siegen, Germany; jens.winter@uni-siegen.de

**Keywords:** piezoelectricity, piezoceramics, actuating materials, lead-free, relaxor, ferroelectrics, BNT–BT, phase transformation, X-ray powder diffraction, self-heating

## Abstract

The electric field response of the lead-free solid solution (1−*x*)Bi_0.53_Na_0.47_TiO_3_–*x*BaTiO_3_ (BNT–BT) in the higher BT composition range with *x* = 0.12 was investigated using in situ synchrotron X-ray powder diffraction. An introduced Bi-excess non-stoichiometry caused an extended morphotropic phase boundary, leading to an unexpected fully reversible relaxor to ferroelectric (R–FE) phase transformation behavior. By varying the field frequency in a broad range from 10^−4^ up to 10^2^ Hz, BNT–12BT showed a frequency-dependent gradual suppression of the field induced ferroelectric phase transformation in favor of the relaxor state. A frequency triggered self-heating within the sample was found and the temperature increase exponentially correlated with the field frequency. The effects of a lowered phase transformation temperature T_R–FE_, caused by the non-stoichiometric composition, were observed in the experimental setup of the freestanding sample. This frequency-dependent investigation of an R–FE phase transformation is unlike previous macroscopic studies, in which heat dissipating metal contacts are used.

## 1. Introduction

Piezoelectric materials exhibit the property of converting electrical energy into mechanical energy and vice versa. This feature is required in a broad range of devices, such as actuators, transducers, sensors, nano-positioners, ultrasonic motors, imaging devices, and other applications [1]. Amongst the piezoceramics, lead-containing compositions are most widely used. About 95% of applied piezoelectric material consists of global, market-dominating, lead-based solid solutions, in particular Pb(Zr_x_Ti_1−x_)O_3_ (PZT) [2,3]. However, lead and lead oxide (PbO) are both found to be hazardous for human health and the environment [4]. Due to the rapidly increasing amount of electrical and electronic waste, the European Waste Electrical and Electronic Equipment Directive (WEEE) / Restriction of Hazardous Substances (RoHS) was adopted by the EU parliament, in order to protect human health and environment from toxic and harmful substances [5]. 

The EU legislation sparked research on lead-free compositions and led to an exponential increase in publications in the last two decades [3,6]. Among several promising materials, bismuth sodium titanate (BNT)-based ceramics have attracted wide interest in the scientific community as a possible alternative. Despite extensive studies, the correlation between macroscopic properties and crystal structure is still not fully understood [7]. The solid solution of BNT with barium titanate (BT), namely (1-x)Bi_0.5_Na_0.5_TiO_3_–xBaTiO_3_ (BNT–xBT), was first mentioned by Takenaka et al. in 1991 [8]. Based on dielectric and piezoelectric property measurements, this study presents a temperature-composition phase diagram with a morphotropic phase boundary (MPB), separating the rhombohedral BNT and the tetragonal BT phase, at compositions with about 6%–7% BT [9]. Similar to PZT, a high dielectric permittivity, electromechanical coupling factor k_33_, and significantly enhanced piezoelectric properties are reported in this region. 

Using in situ X-ray diffraction methods, Jo et al. showed that upon electrical poling, the MPB extends to a broader range of 6%–11% BT [1]. Vakhrushev et al. [10] inferred rhombohedral clusters that are surrounded by a weakly polar tetragonal matrix around the MPB in pure BNT. Later, a more detailed model was published by Kreisel et al. [11], with planar polar defects in an otherwise rhombohedral matrix. The different behavior of coexisting phases could also be confirmed by in situ experiments on BNT-based compositions by Hinterstein et al. [9,12]. These findings indicate an analogy to “relaxor ferroelectrics”, a designation made common by Cross [12,13,14]. A relaxor ferroelectric is characterized by a frequency dispersion in the temperature-dependent dielectric permittivity and a comparatively broad maximum [15,16]. Works based on diffraction suggest that the relaxor state in the MPB composition range, comprises a phase with slight non-cubic distortions, a so-called pseudo-cubic phase [17,18,19]. 

On the tetragonal side of the MPB for BT contents above 11%, the pseudo-cubic phase is reduced and relaxor properties vanish with increasing BT content, in favor of a predominantly tetragonal symmetry with ferroelectric properties [8,20,21,22]. A compositional phase diagram of BNT–xBT, suggested by Zhou et al. is depicted in Figure 1 [20]. 

The high BT composition range near the MPB (~12%) is comparatively rarely investigated, unlike the MPB and BT-rich (>14%) compositions, in which valuable properties have been observed [23]. Among them, the already commercially available, lead-free, BNT-based PIC 700 serves as an example. Its high coupling factor of the thickness mode of vibration and a low planar coupling factor are desirable for transducer applications [23]. PIC 700 exhibits a tetragonal unit cell, analogous to the common lead-containing commercial piezoelectric PIC 151 (PI Ceramics, Lederhose, Germany) [24]. 

By externally applying an electric field to the relaxor (R) state, BNT–BT undergoes a phase transformation to the ferroelectric (FE) state, which can be macroscopically indicated by a double polarization hysteresis loop [17,19]. In the particular case of a field-dependent reversibility between the R and FE state, high macroscopic strain values are expected and are, therefore, attractive for actuator applications [3]. Apart from the amplitude of the electric field, the frequency and the temperature affect the strain in both lead-containing and lead-free actuator applications [25,26]. Highly challenging conditions for a stable functionality have to be fulfilled, since a broad range of frequencies and a maximum range of working temperatures are desired [27]. The applied frequency is limited not only by operational conditions such as the electrical driver power, but also by the resonant frequency of the mechanical system and self-heating of the actuator [28]. In the case of fuel injectors, the challenge lies in the substantial amount of self-generated heat of PZT stack actuators, at high frequencies and field amplitudes that are usually required [25]. Self-heating generally leads to degradation and affects the lifetime, durability and the positioning accuracy of the actuator and is caused by mechanical and dielectric loss, which can be related to the piezoelectric displacement hysteresis [25,29,30]. Another impact of the frequency is found by previous works investigating both lead-containing and lead-free materials, showing an exponential decrease of the coercive field, with higher frequencies [26,31,32,33]. 

Beside the stoichiometric BNT, non-stoichiometric compositions were investigated in previous works by Li et al. in regard to their microstructure and electrical properties [34,35]. Bi and Na occupy the A-site and Ti occupies the B-site in the perovskite structure ABO_3_ of BNT. The authors found significant differences in conductivity by varying both the ratios of A-/B-site and Bi/Na, within the A-site. The stoichiometric, Na-excess, and Bi-deficient compositions of BNT led to conducting properties, whereas Bi-excess and Na-deficiency yielded insulating materials. These findings occurred regardless of A-site excess or deficiency. Although a relaxor state is not expected for stoichiometric compositions with higher BT content, the introduced non-stoichiometric Bi-excess, however, significantly lowers the temperature of the R–FE phase transformation T_R–FE_ [1,3,36,37,38]. Acosta et al. demonstrated for BNT–25ST that an increasing temperature suppresses the field induced phase transformation [27]. This is due to the high temperature that increases the energy barrier of a ferroelectric order, thus, requiring higher fields for an R–FE phase transformation. Accordingly, the authors determined a negligible amount of the ferroelectric state above 110 °C [27]. 

Hence, the R–FE phase transformation in BNT–BT is not only influenced by the stoichiometry and composition, but also by the temperature that is directly affected by the electric field frequency. To understand and enhance the desired strain properties for technical applications, the exact and comprehensive influences of all these aspects require detailed consideration.

The composition investigated in the present work combines the two characteristics, a high BT content (>11%) and a non-stoichiometry, in order to achieve a highly reversible field induced R–FE phase transformation. In situ electric-field-dependent transmission measurements performed with synchrotron X-ray radiation allow to detect and quantify the structural bulk response. Due to the special experimental setup of a freestanding sample, the frequency and temperature-dependent R–FE phase transformation can also be investigated without significant heat dissipation.

## 2. Materials and Methods 

### 2.1. Sample Processing and Preparation

Bi_0.5_Na_0.5_TiO_3_–12BaTiO_3_ was processed using the conventional solid-state route. The precursor powders (Alfa Aesar, Karlsruhe, Germany) Na_2_CO_3_ (99.5%), Bi_2_O_3_ (99.95%), BaCO_3_ (99.8%), and TiO_2_ (99.9%) were used. The powders were weighed with 2% bismuth excess. Extra Na_2_CO_3_ was added to compensate for moisture, instead of drying Na_2_CO_3_ and weighing. First, the powders were mixed in a ball mill in pure ethanol with yttria-stabilized zirconia media (Tosoh, Tokyo, Japan) for 24 h. The slurry was dried overnight at 80 °C, in an oven. The powder was calcined at 900 °C for 4 h in closed alumina crucibles, crushed, and milled for 96 h and then dried again. Green bodies of samples with a diameter of around 8 mm and a thickness of about 2 mm were prepared by uniaxial pressing, followed by cold isostatic pressing (ISA-CIP-60-100-400-AL, Ilshin Autoclave Co., Ltd., Daejeon, Korea), at a pressure of 415 MPa to densify the pellets. The pellets were sintered at 1150 °C for 3 h, in alumina crucibles with lids. Rectangular bar shaped samples of approximate dimensions 1 × 1 × 5 mm^3^ were cut and polished from sintered pellets. Silver conductive paint was manually applied as electrodes on two opposite long sides of the samples. The chemical composition of BNT–12BT was analyzed using inductively coupled plasma optical emission spectrometry analysis (ICP–OES).

### 2.2. Diffraction Experiments

BNT–12BT samples were measured at beamline P02.1 at the Deutsches Elektronen-Synchrotron (DESY) in Hamburg, Germany [39,40]. This beamline provides high-energy X-rays with a photon energy of around 60 keV, with a wavelength of 0.20718 Å. Using a specially tailored sample environment setup, the diffraction experiments were performed in transmission geometry, in order to characterize the bulk properties. The sample environment (Appendix A) is described in greater detail in the Appendix A. A two-dimensional (2D) flat panel detector enabled data acquiring with texture effects from the field induced sample response, during field cycling. The 16-inch detector of the XRD 1621 N ES Series (PerkinElmer) with 2048 × 2048 pixels and a pixel size of 200 µm^2^ collected the diffraction images. An exposure time of 30 s yielded satisfactory statistics for every diffraction image. For in situ electric-field-dependent measurements, an external electric field of 5 kV/mm was applied to the sample, with the electric field vector placed perpendicular to the incident beam direction. For an entire bipolar electric field cycle, 161 diffraction images of 30 s were acquired using stepwise changing field values, resulting in an effective frequency of 1.9 × 10^−4^ Hz.

In order to measure at higher frequencies (1.0 × 10^−3^ Hz and 5.0 × 10^−2^ Hz), exposure times inevitably fall below the limit of 1 s per image required for the stepwise-field acquisition mode. To obtain sufficiently high signal-to-noise ratios, the 2D detector operates in the high-gain setting, while measuring in continuous wave electric field mode.

To further increase the frequencies to 10^0^, 10^1^, and 10^2^ Hz, the 1D multi-analyzer point detector (MAD) [41] is instead used for data collection in the stroboscopic mode [42,43]. In contrast to the 2D detector, no texture effects can be measured simultaneously, since the diffraction patterns are collected by the detector for only one specific azimuthal angle. In exchange, a significantly higher angular resolution is possible. The MAD detector consists of ten data acquiring channels, which simultaneously detect ten different specific 2θ ranges of the diffraction pattern. In this case, the ranges are chosen to completely cover the 111*_C_* and 200*_C_* reflections. By splitting the electric field pulse into sub-second time channels, the stroboscopic mode allows data collection with high counting statistics, however high the field frequency. Details about the measurement can be found in Appendix A section (Appendix A). Even though two different detectors were used, all experiments were conducted at the same beamline P02.1, with identical conditions.

The temperature-dependent measurement of BNT–12BT was carried out at beamline ID15A of the European Synchrotron Radiation Facility (ESRF) in Grenoble, France [44]. The monochromatic beam with a photon energy of 86 keV and a cross-section (H × V) of 150 × 100 µm^2^ was used in transmission geometry. A Pilatus 3X CdTe detector (1679 × 1475 pixels, pixel size 172 × 172 µm^2^) was used for data collection. For this measurement, an unpoled sample of BNT–12BT was used. The diffraction pattern at room temperature and patterns starting at 54 °C until 115 °C were collected in 1 °C steps.

### 2.3. Data Analysis

The images acquired by the 2D detector were divided into 18 azimuthal sectors (“cakes”) of 5° angle widths and each were integrated using the software Fit2D to obtain intensity against 2θ values. For the frequency-dependent measurements at 10^0^, 10^1^, and 10^2^ Hz, acquired by the MAD detector, the diffraction patterns during the applied maximum field and in the remanent state were analyzed. This was done by Rietveld refinement of the ten individual measurement ranges, combined to a single pattern. The temperature-dependent diffraction data at room temperature and from 54 °C to 115 °C collected at ESRF were integrated over the whole 90° azimuthal section. The obtained patterns were all refined with the Rietveld method, using the software package Material Analysis Using Diffraction (MAUD) [45]. The refinement details are described in the Appendix A section (Appendix A). To compare the results with the macroscopic measurements, a thin film analyzer (TF Analyzer 2000, aixACCT Systems GmbH, Aachen, Germany) was used for electromechanical characterization, using electric field frequencies from 10^−2^–10^1^ Hz.

## 3. Results and Discussion

An ICP–OES analysis of the BNT–12BT sample (Table 1) yielded the detailed stoichiometric composition as 0.88Bi_0.53_Na_0.47_TiO_3_–0.12BaTiO_3_. Starting from the mass percentages, the molar percentages were calculated using the molar masses of bismuth, sodium, barium, titanium, and zirconium, respectively. The contamination with zirconium resulted from grinding using yttria-stabilized zirconia media. Normalizing to titanium as one mol, the other molar percentages were divided by the molar percentage of titanium. The normalized amount of Barium yielded 0.122(1) mol and Bi and Na yielded 0.878(1) mol, which led to the approximate indication of 0.12BT and 0.88BNT. Regarding 0.878(1)BNT, the amount of 0.467(2) Bi and 0.411(3) Na resulted in approximately 0.53 mol Bi and 0.47 mol Na. This led to a composition of 0.12 BaTiO_3_–0.88 Bi_0.53_Na_0.47_TiO_3_.

In previous works, the crystallographic phase of BNT–12BT was identified as tetragonal (P4*mm*) in the ferroelectric state, at room temperature [1,21,38,46,47]. However, even small A-site off-stoichiometric compositional deviations, such as bismuth excess or deficiency, can significantly influence the dielectric and piezoelectric behavior. Due to the high Bi_2_O_3_ volatility during sintering, bismuth vacancies $\left(V_{Bi}^{\prime \prime \prime }\right)$ can be usually expected, which lead to higher concentrations of oxygen vacancies $\left(V_{O}^{••}\right)$ [48]. Impurities are mostly present as acceptor dopants and give rise to a similar effect. Introducing bismuth excess compositions, Li et al. observed that fewer bismuth vacancies and increasing amounts of ($Bi_{Na}^{••}$) defects reduced the concentration of oxygen vacancies [35]. This is desirable, since a low amount of oxygen vacancies should not only inhibit conductivity, but should also stabilize the long-range ferroelectric state and impede the relaxor state. The temperature of the relaxor to ferroelectric state transformation T_R–FE_ can thereby be lowered [38]. As shown in the following Kröger-Vink equation, bismuth excess shifted the equilibrium to the left side and reduced the concentration of bismuth and oxygen vacancies [34,35,49]:(1)$$2Bi_{Bi}^{\times }+3O_{O}^{\times }⇄2V_{Bi}^{\prime \prime \prime }+3V_{O}^{••}+Bi_{2}O_{3}$$

Using the BNT–12BT sample, macroscopic field-dependent strain (Figure 2a) and polarization (Figure 2b) measurements were performed with the aixACCT instrument, using different frequencies and a field amplitude of 5 kV/mm.

The polarization curves exhibit a double-loop behavior, which is characteristic for electric field induced R–FE phase transformations [17,19]. Since the double loop can be seen at all frequencies, it can be assumed that the phase transformation occurs in all of these cases.

To gain a more detailed insight, the BNT–12BT sample, both with and without an applied electric field were investigated by in situ synchrotron diffraction. Additionally in situ diffraction yielded information on texture, strain, and its evolutions, as a function of the external field and its frequency. Figure 3 depicts selected diffraction patterns of the BNT–12BT sample during one bipolar electric field cycle with an amplitude of 5 kV/mm and a frequency of 1.9 × 10^−4^ Hz. The characteristic 111*_C_* and 200*_C_* reflections of the diffraction patterns are depicted enlarged. Throughout this work, the reflections are indexed with the subscripts *C* and *T,* which denote the cubic (*Pm*$\bar{3}$*m*) and tetragonal (*P4mm*) perovskite unit cells, respectively. By azimuthal slicing of the 2D diffraction pattern, the contributions of the differently oriented domains can be differentiated. The azimuthal angles φ = 0°, 45°, and 90° of the plotted patterns, correspond to the conditions in which the scattering vector is parallel, 45°, and perpendicular to the electric field vector.

For the unpoled sample prior to electrical loading, no texture effects exist (Figure 3a). In the case of the remanent state, the different orientations show no difference to the unpoled state, which means that the field-induced response vanishes fully. A reversible phase transformation between the R and the FE states can therefore be inferred, since the splitting occurs only with application of the electric field and ceases in the remanent state. This reversible behavior remains even after 10^4^ electric field cycles, which is shown in Appendix A of the Appendix A section. Figure 3b shows the electric field induced phase transformation and evolution of texture effects resulting from non-180° domain switching processes and lattice strain in the applied field state at 5 kV/mm. The electric field induced R–FE phase transformation and the texture effects are visible especially at the 200*_C_* reflection. The response at maximum field is illustrated in greater detail in Figure 4.

The 111*_C_* reflection (Figure 4a) shows a shift to higher 2θ with larger azimuthal angles φ, which is attributed to the effect of the lattice strain (black arrow) [50,51]. A more pronounced field-induced response can be observed for the 200*_C_* reflection (Figure 4b) as it splits into distinguishable reflections, due to the R–FE phase transformation and non-180° domain switching [50,51]. Since the field-induced splitting only occurs in the 200*_C_* rather than in the 111*_C_* reflection, a field-induced tetragonal phase evolution can be corroborated. It should be noted that the 002*_T_* and the 200*_T_* reflections remain at the same 2θ positions, due to domain formation and switching instead of lattice strain (black arrows). This is in accordance with previous works done with PZT and BNT–7BT [17,51]. The increasing intensities of 002*_T_* and decreasing intensities of 200*_T_* with lower angles φ indicate a more facile domain formation for the grains, whose 002*_T_* vector is oriented parallel to the electric field vector. 

The third reflection of the 200*_C_*triplet at 2θ = 6.05°, between the two tetragonal split reflections, indicates the occurrence of a pseudo-cubic second phase [33,52]. A similar behavior was observed for the relaxor system BNT–25ST, in which a second phase was also present [53]. The fact that the reflection at 2θ = 6.05° shifted as a function of φ (red arrow) indicates lattice strain in the second phase. It remains unclear whether or not this reflection is pseudo-rhombohedral, although according to previous studies, this phase exhibits a pseudo-rhombohedral character in similar materials [9,12,53]. 

Different frequencies ranging from 1.9 × 10^−4^ Hz to 10^2^ Hz of the bipolar electric field were applied with the same amplitude of 5 kV/mm. Since the 200*_C_* reflection illustrates the tetragonal phase transformation, the course of one bipolar field cycle of the 200*_C_* reflection is plotted in Figure 5 as a function of time. Lower frequencies were measured with the 2D detector (Figure 5a–c), while high frequency in situ measurements were carried out in stroboscopy mode, using the 1D high-resolution MAD detector (Figure 5d–f) [42]. The higher angular resolution, therefore, explained the smaller reflection widths in d–f.

For all field frequencies, the sample showed a reversibility upon applied field. At lower frequencies (Figure 5a–c), a striking smearing out of the 200*_C_* reflection was clearly discernible, which could be ascribed to the reversible pseudo-cubic to tetragonal phase transformation, upon applying field. Increasing the electric field, the center of reflection first remained at the same 2θ positon and then shifted to lower 2θ values. The shifting was caused by intrinsic lattice strain effects. Additionally, a splitting at maximum field value remained to a small extent at high frequencies (d–f). 

Figure 6 depicts the 200*_C_* reflections at the maximum applied field (a) and at the remanent field state (b), for all frequencies. Additionally, the reflection at 0 Hz in Figure 6a (red line) represents a static field measurement. At maximum field, the field induced tetragonal 200*_C_* reflection splitting diminishes by applying higher frequencies (black arrows). This is first identified by the decreasing intensities of the tetragonal 002*_T_* and 200*_T_* reflections, which indicate decreasing tetragonal phase fractions. Secondly, the 2θ distance between the split tetragonal reflections decreases and coincides until 10^2^ Hz. Hence, simultaneous to the decreasing phase fraction, the lattice distortion of the tetragonal unit cell also decreases.

The most pronounced splitting occurs at static field acquired by the high-resolution detector, followed by still clearly visible splitting at frequencies in the range 10^−4^–10^-2^ Hz, measured by the 2D detector. In the range of 10^0^–10^2^ Hz, measured stroboscopically with the high-resolution detector, the splitting is barely discernible. However, the reflections are significantly broader than in the remanent state (Figure 6b). It can, therefore, be concluded that the apparent splitting is indeed frequency-dependent rather than a consequence of the choice of the detector.

In Figure 6b, the reflections shift towards smaller 2θ values with increasing frequencies, which cannot be attributed to an instrumental zero shift, even though two different detectors were used. However, a temperature increase due to self-heating might be the reason for this observation. In order to verify this assumption, temperature-dependent synchrotron measurements of an unpoled BNT–12BT sample at the ESRF synchrotron source were performed and the diffraction patterns were refined using the Rietveld method with a cubic (*Pm*$\bar{3}$*m*) phase, to determine the linear expansion coefficient α. The temperature-dependent change of the cubic lattice parameter is plotted in Figure 7a. The thermal expansion showed an unambiguously linear course and can be fitted with Equation (2), which demonstrates the unit cell parameter dependence on the temperature.
(2)$$a=\alpha \cdot T+a_{0},$$
where α = 45.1(2) × 10^−6^ K^−1^ is the linear expansion coefficient and $a_{0}$ = 3.91583(2) Å the unit cell parameter at 0 °C. The values in parentheses refer to the calculated errors. The diffraction patterns of the remanent state at frequencies of 1.9 × 10^−4^, 10^0^, 10^1^, and 10^2^ Hz were refined in the same way to yield the unit cell parameters for all frequencies. An effect of unit cell expansion is concluded, thereupon, which can be caused by an increasing temperature.

To determine the temperatures for the different frequencies of the field-dependent measurements, the linear expansion coefficient was considered together with the determined unit cell parameters. Assuming a possible temperature effect in the field-dependent measurements, the unit cell parameters from the Rietveld refinement were inserted into the linear equation (Equation (2)) to calculate the temperature. This yielded a frequency-temperature plot (Figure 7b) fitted with an exponential function:(3)$$T=b\cdot f^{c}+d,$$
where the pre-exponential factor b = 25(2) Ks^b^, exponent c = 0.27(1), and intercept d = 20(2) K. Accordingly, a frequency-dependent field induced self-heating within the sample due to power dissipation during the hysteresis, is being supposed [27,54].

The decreased tetragonal phase fraction for elevated temperatures is qualitatively explicable in terms of field-temperature phase diagrams of several similar relaxor and ferroelectric compositions based on temperature-dependent structural measurements [38,55,56,57]. The results suggest that the course of the phase transformation values is U-shaped in the electric field-temperature diagram—with increasing temperature, T_R-FE_ first decreases to lower electric fields and after the minimum is reached, it increases exponentially [38,58]. Hence, moving to higher temperatures, the FE long-ranged order cannot be reached by applying an electric field of amplitude 5 kV/mm.

The split 002*_T_* and the 200*_T_* reflections coincide almost at the field frequency of 10^0^ Hz, which indicates the prevailing relaxor state with an almost reduced FE state (Figure 6). The frequency of 10^1^ Hz corresponds to a temperature of about 45 °C, according to the exponential temperature fit (Figure 7b), where the electric field does not suffice for the R–FE phase transformation. This T_R-FE_ temperature threshold is substantially lower than that found in the investigated lead-free materials in the literature [26,27,38,55,56,57]. In the case of BNT–12BT studied by Shi et al., T_R-FE_ of ~205 °C is found for the electric field of 5 kV/mm [38]. As a possible reason for the huge deviation of the T_R-FE_ temperature thresholds, the experimental setup of macroscopic measurements in general can be taken into account, since comparatively large metal fixing contacts are used herein, which directly dissipate possibly self-generated heat from the sample. This means that in macroscopic measurements the heat dissipation is different from the freestanding sample setup in silicon oil for in situ synchrotron experiments. Therefore, special care has to be taken for intermediate frequencies, when comparing structural with macroscopic studies.

Apart from the temperature effect, other high-frequency triggered effects should be taken into account for the suppressed phase transformation. This is especially evident from clearly noticeable changes in 200*_C_* reflection splitting from the low frequency range of 10^−4^–10^−2^ Hz in Figure 6, while no substantial temperature increase happens according to the temperature-dependent measurement. One effect is supposedly owed to kinetics. It was shown for PZT that for very low frequency, the creep effects exist [33,59]. With decreasing frequency, the macroscopic strain (Figure 2a) increased and in the same course the unit cell distortion as well as the tetragonal phase fraction occurred. This can be seen very well in Figure 6a, marked by the arrows. After the formation of the FE state in BNT–12BT, the unit cell distortion increased with time, due to the creep effects during the applied field. At higher frequencies, the sample remained in the pseudo-cubic state, since the domains in the FE long-range order were inhibited and eventually their formation prevented due to a lack of time, to build up and reorient.

## 4. Conclusions

The results showed that BNT–12BT on the tetragonal side of the MPB with higher BT compositions exhibited a reversible relaxor to ferroelectric (R–FE) phase transformation, upon applied electric field. This phase transformation behavior can be regarded as an extended MPB, induced by introducing Bi-excess for a non-stoichiometric composition. A field-dependent R–FE phase transformation is desirable, owing to the usually accompanied large macroscopic strain effects [3]. We found that the field frequency is of vital importance, as for the elevated frequencies, the R–FE phase transformation was suppressed and the sample remained in the pseudo-cubic state. High-field frequencies triggered an internal self-heating within the sample, impeding the creation of the long-range ordered ferroelectric state. The self-heating phenomenon was substantially less discernible in macroscopic measurement setups, since large, sample-fixing metal contacts were used, which extract the heat from the sample. Hence, the method using field-dependent in situ X-ray data sheds light on the internal temperature effects of samples under zero-stress conditions. Besides the self-heating at higher frequencies, kinetic effects led to a suppressed formation of the long-range ordered ferroelectric state and, therefore, additionally impeded the R–FE phase transformation. Based on an extraordinary broad range (10^6^) of applied field frequencies, this work provides essential insights into the structural behavior of the promising lead-free alternative material BNT–BT. This knowledge contributes to the development and optimization of future lead-free piezoceramics and stimulates further work for commercialization of applicable functional ceramic systems.

## Figures and Tables

**Figure 1 materials-13-01054-f001:**
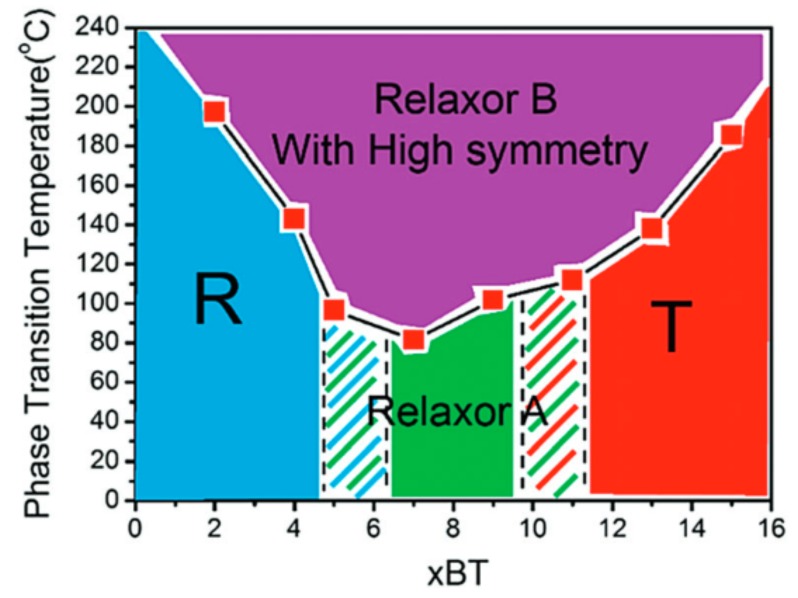
Phase diagram of (1−*x*)Bi_0.53_Na_0.47_TiO_3_–*x*BaTiO_3_ (BNT–*x*BT) by Zhou et al. Reproduced from Ref. [20] with permission from the PCCP Owner Societies.

**Figure 2 materials-13-01054-f002:**
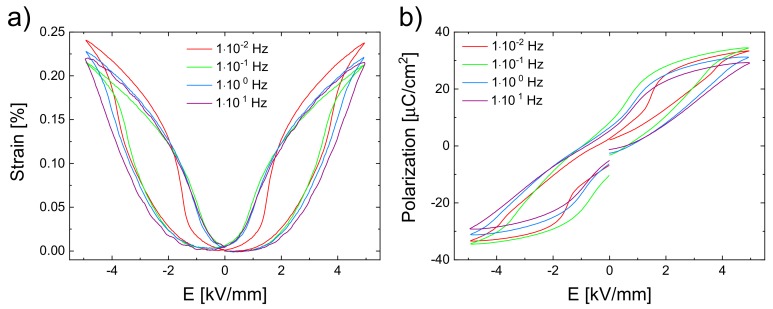
Macroscopic (**a**) strain and (**b**) polarization curves of the BNT–12BT sample at different frequencies resulting from one bipolar cycle of an applied electric field of 5 kV/mm.

**Figure 3 materials-13-01054-f003:**
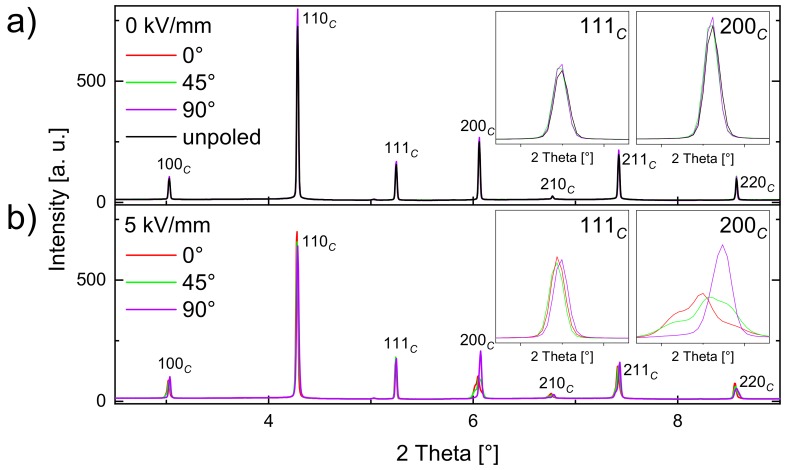
Orientation-dependent diffraction patterns of BNT–12BT in the remanent state (**a**, red, green, and blue line) and one orientation of the unpoled virgin state (**a**, black line). Since the patterns are almost identical, the colored lines mostly lie on top of each other. Maximum electric field at 5 kV/mm (**b**) during one bipolar cycle at a frequency of 1.9–10^−4^ Hz. Three different azimuthal angles φ = 0°, 45°, and 90° are shown, which represent scattering vectors that lie parallel, 45°, and perpendicular to the electric field vector, respectively. The insets show the 111*_C_* and 200*_C_* reflections, which are considered to indicate tetragonal or rhombohedral unit cell distortions and domain orientations.

**Figure 4 materials-13-01054-f004:**
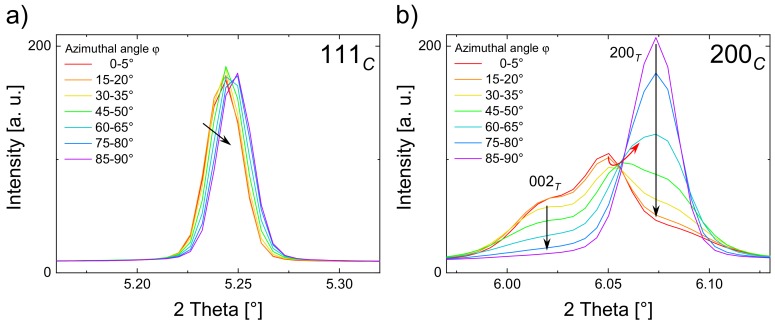
Orientation-dependent diffraction pattern of the 111*_C_* reflection (**a**) and the 200*_C_* reflection (**b**) at different azimuthal angles φ in the applied field state at 5 kV/mm.

**Figure 5 materials-13-01054-f005:**
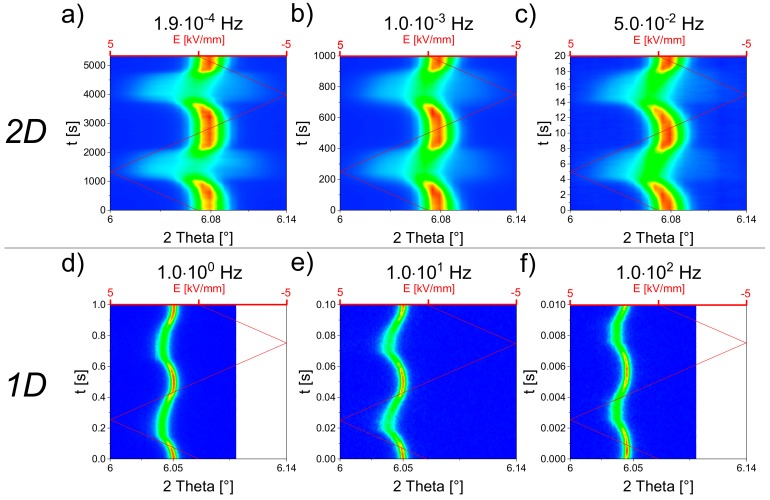
Electric-field-dependent contour plots of the 200*_C_* reflection of BNT–12BT with different field frequencies. The colors denote intensity values, where red represents the highest intensity. Above (**a**–**c**): Intensity plots measured at the azimuthal angle 0° from the 2D detector. Below (**d**,**e**,**f**): Intensity plots obtained from the 1D (azimuthal angle 0°) high-resolution MAD detector recorded in the stroboscopy mode. No intensity data were collected at the whitened region in (**d**) and (**f**).

**Figure 6 materials-13-01054-f006:**
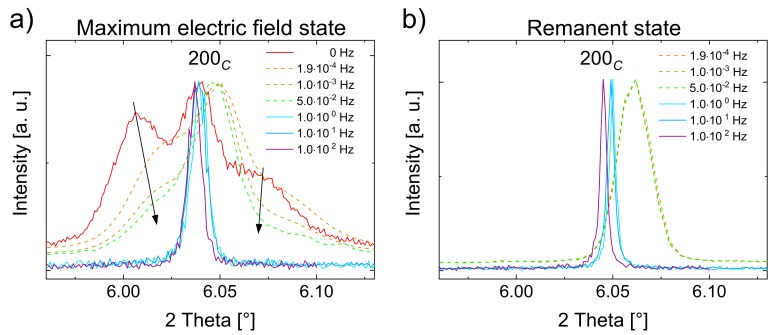
High-resolution (solid line) and 2D detector (dotted line, all at 0° azimuthal angle) patterns of the 200*_C_* reflection at maximum electric field during bipolar cycling (**a**) and in the remanent state (**b**) at different frequencies from static field (0 Hz) to 10^2^ Hz. To be graphically comparable, the intensities were normalized to the maximum intensity.

**Figure 7 materials-13-01054-f007:**
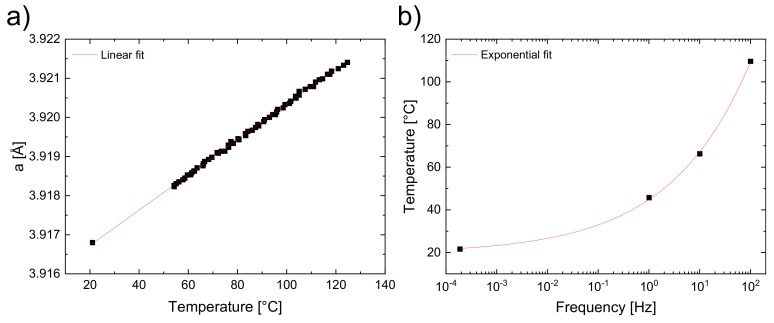
Temperature-dependent measurement of BNT–12BT with linear fitted unit cell parameters obtained from the Rietveld refinement (**a**). Experimental data and exponential temperature fit $T=b\cdot f^{c}+d$ of different field frequencies 1.9 × 10^−4^, 10^0^, 10^1^, and 10^2^ Hz (**b**).

**Table 1 materials-13-01054-t001:** Inductively coupled plasma (ICP) analyzes of BNT–12BT.

Element	Bi	Na	Ba	Ti	Zr
**ICP Data (wt. %)**	42.6(1)	4.12(3)	7.33(4)	20.89(5)	0.1115(5)
**Element Amount (mol)**	0.467(2)	0.411(3)	0.122(1)	1	0.00280(1)

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
