# Peer review of "Electric-Field-Induced Phase Transformation and Frequency-Dependent Behavior of Bismuth Sodium Titanate–Barium Titanate"

_materials, 2020, doi:10.3390/ma13051054_

Round 1

Reviewer 1 Report

This paper reports the findings from in situ synchrotron XRD experiments on BNT-0.12BT. Overall, this paper is well written and clearly presented, and logical conclusions are drawn from the presented results. However, the references to the figures in the text haven't compiled correctly. Whilst I could work out which figures were being referred most of the time, it becomes more difficult in the last two paragraphs of the results and discussion section (page 10, line 337 onwards). I would like to re-read this section once the references in the text are showing correctly before I can recommend the paper for publication.

There are a couple of other minor queries that should also be addressed, listed below.

ICP-OES analysis should be defined before using the acronym. 

The colours of the line plots are very faint on Figure 2. Could perhaps the thickness of these lines be increased to help the reader distinguish between them. 

On page 6, line 235, the authors state that the 'reversible behaviour remains even after 10^4 cycles.' Is there any experimental evidence for this claim?

On page 10, line 332, numbers are given in brackets after the fit parameters from the exponential model. Do these need to be there and if so, could the authors state the relevance?

For a reader unfamiliar with the BNT-BT system, a phase diagram would be useful for reference in the introduction. 

Reviewer 2 Report

The manuscript describes the electric filed induced phase transformation (R-FE) of BNT-12BT composition, which is analysed by the in-situ synchrotron X-ray powder diffraction. Also investigated the frequency dependent dielectric properties and strain vs electric field loop analysis. Authors executed the problem statement in systematic way and discussed the corresponding scientific discussions in fruitful manner. In my opinion, the proposed manuscript is suitable for publication after the completion of minor queries.

  1. HR-TEM and HAADF-STEM measurements expose the phase distribution and intrinsic lattice strain information of BNT-12BT sample at atomic sites.
  2. In my opinion, in-situ electric field dependent diffraction studies alone not enough to confirm the modulation in the tetragonal state of proposed compound. Authors should provide the field dependent angular Raman scattering of the BNT-12BT, it will give you a complete analysis of phase dependent active phonons modes.
